

# Signal region combination with full and simplified likelihoods in MADANALYSIS 5

Gaël Alguero[1,2]⋆, ⬤ Jack Y. Araz[3]†, ⬤ Benjamin Fuks[4]‡ and ⬤ Sabine Kraml[1]°

**1** Laboratoire de Physique Subatomique et de Cosmologie (LPSC), Université Grenoble-Alpes, CNRS/IN2P3, 53 Avenue des Martyrs, F-38026 Grenoble, France
**2** LAPTh, Univ. Grenoble Alpes, USMB, CNRS, F-74940 Annecy, France
**3** Institute for Particle Physics Phenomenology, Durham University, South Road, Durham, DH1 3LE
**4** Laboratoire de Physique Théorique et Hautes Energies (LPTHE), UMR 7589, Sorbonne Université et CNRS, 4 place Jussieu, 75252 Paris Cedex 05, France

⋆ gael.alguero@lpsc.in2p3.fr , † jack.araz@durham.ac.uk ,
‡ fuks@lpthe.jussieu.fr , ° sabine.kraml@lpsc.in2p3.fr

## Abstract

The statistical combination of disjoint signal regions in reinterpretation studies uses more of the data of an analysis and gives more robust results than the single signal region approach. We present the implementation and usage of signal region combination in MADANALYSIS 5 through two methods: an interface to the PYHF package making use of statistical models in JSON-serialised format provided by the ATLAS collaboration, and a simplified likelihood calculation making use of covariance matrices provided by the CMS collaboration. The gain in physics reach is demonstrated 1.) by comparison with official mass limits for 4 ATLAS and 5 CMS analyses from the Public Analysis Database of MADANALYSIS 5 for which signal region combination is currently available, and 2.) by a case study for an MSSM scenario in which both stops and sbottoms can be produced and have a variety of decays into charginos and neutralinos.



# 1 Introduction

Searches for new physics at the LHC are typically performed in specific bins of kinematic distributions, so-called signal regions (SRs), designed to maximise the number of events from the hypothesised signal with respect to the number of "background" events originating from Standard Model processes. In parallel, control and validation regions are defined in the phase space where no or very little signal from new physics is expected. A statistical analysis is then performed to evaluate the confidence level of the hypothesised Beyond the Standard Model (BSM) scenario, and claim evidence for or set a limit on the new particles of this scenario.

Reinterpretation studies [1,2] outside the experimental collaborations, like achieved with MADANALYSIS 5 [3], aim at reproducing this process for BSM scenarios different from those considered in the original experimental publication. This makes it possible for the community as a whole to test a much larger variety of theories against LHC results than would be possible purely within the experimental collaborations. Moreover, it enables phenomenologists to pursue global analyses, give detailed feedback on the physics impact of the experimental results, and suggest target BSM scenarios for future investigations. An overview of publicly available reinterpretation tools, together with an extensive discussion on which information is needed from the experimental collaborations, is given in [2].

Among the essential information figure the statistical models used to derive results in experimental analyses [4]. In this context, a long-standing problem for reinterpretation studies consists of the statistical combination of disjoint SRs, as the information on the correlation of uncertainties is not always provided by the experimental collaborations. In the absence of appropriate correlation information, it is common practice to use only the SR with the *best expected* sensitivity to the BSM parameter point under investigation (colloquially called the "best SR") for the statistical evaluation, *e.g.* for limit setting. The reason is that the choice of SR, and thus the conclusions drawn from data, must not depend on statistical fluctuations which affect observed data. This entails a number of problems. First of all, in the best-SR approach only a part (sometimes a very small fraction) of the available data is used, which can lead to false conclusions. Typically this causes a loss in sensitivity, but, as we will see, it can also lead to too strong exclusions. Furthermore, if the best SR changes from point to point in a scan, this can lead to numerical instabilities in global fits. We refer to [4] and references therein for more discussion of these and related issues.

The CMS collaboration provides correlation information for some of their searches for supersymmetry (SUSY) and other new particles in the form of approximate covariance matrices, designed to build a so-called *simplified likelihood* [5]. The underlying assumptions are that systematic uncertainties in the signal modelling can be neglected, and that uncertainties on the background contributions are Gaussian in shape (implying that the distribution of the number of background events is symmetric around the expectation). Although approximate, the combination of SRs through this simplified likelihood scheme greatly improves the precision and constraining power of analysis recasts relative to the usage of the best SR only.[1]

The ATLAS collaboration follows a different strategy: instead of approximate SR correlations, the collaboration recently started to provide *full statistical models* in JSON-serialised format [7]. These statistical models, based on HISTFACTORY [8], describe the complete probabilistic dependence of the observable data on both the parameters of interest and the nuisance parameters. When the observed numbers of events are entered, this becomes the likelihood function (see [4] for details). The JSON-serialised format enables the usage of the HISTFACTORY structure outside the ROOT framework, in particular within the PYHF package [9,10]. PYHF can conveniently be used for signal patching, evaluation of likelihoods, computation of

---

[1]Non-Gaussian effects, which can become important for instance when uncertainties are systematics dominated, could be accommodated in an extended simplified likelihood framework as proposed in [6].

Table 1: Overview of MADANALYSIS 5 recast codes for which SR combination is available. The last column specifies the statistical information used for the combination. Here, "full (simplified) model" stands full (simplified) JSON-serialised HISTFACTORY model from ATLAS, to be used with PYHF, while "cov. matrix" stands for covariance matrix in the simplified likelihood approach of CMS. Details are given in section 3.

| Analysis ID | Short description | # SRs | Statistical information |
|---|---|---|---|
| ATLAS-SUSY-2018-04 | Stau search, 2 taus | 2 | full model, all SRs |
| ATLAS-SUSY-2018-06 | EW-inos, 3 leptons | 2 | simplified model, all SRs |
| ATLAS-SUSY-2019-08 | EW-inos, $WH(\to b\bar{b})$ | 9 | full model, all SRs |
| ATLAS-SUSY-2018-31 | Multi-$b$ sbottom search | 8 | full model, 3+3+1 SRs |
| CMS-SUS-16-039 | EW-inos, multi-lepton | 158 | cov. matrix, 44 SRs |
| CMS-SUS-16-048 | OS soft leptons | 21 | cov. matrix, 12+9 SRs |
| CMS-SUS-17-001 | Stops, 2 OS lept. | 3 | cov. matrix, 3 SRs |
| CMS-SUS-19-006 | Multi-jet gluino/squarks | 174 | cov. matrix, all SRs |
| CMS-EXO-20-004 | Multi-jet DM search | 66 | cov. matrix, all SRs |

CL$_s$ values, *etc.* Moreover, it also permits to prune a full statistical model and derive a simplified version of it. The SIMPLIFY [11] tool, for instance, takes a given full statistical model encoded in the JSON format and derives a simplified one in which all nuisance parameters are combined into a single one, and in which all contributions to the background are merged. Such simplified statistical models often (but not always) yield equivalent results for a much smaller computing time, as will be exemplified below.

It is now the task of the public reinterpretation tools to make use of this information. Statistical models from ATLAS and covariance matrices from CMS are already incorporated in SMODELS [12, 13] in the context of re-using simplified-model results. In this paper, we present their implementation and usage in MADANALYSIS 5 for the purpose of full analysis recasts.[2] We first explain in section 2 the technical implementation through the extension of the MADANALYSIS 5 .info XML files pertaining to each analysis. In section 3 we present the analyses for which SR combination is currently available (see table 1), comparing the limits obtained in the best-SR, simplified likelihood and/or full likelihood approaches. In section 4 we illustrate the gain in physics reach by means of a concrete example within the Minimal Supersymmetric Standard Model (MSSM). Section 5 contains our conclusions.

## 2 Technical implementation

In this section we summarise how statistical models in JSON-serialised format and covariance matrices provided for a simplified likelihood treatment can be used in the MADANALYSIS 5 framework. The functionality of SR combination can be turned on/off from the code's command line interface via the command:

```
set main.recast.global_likelihoods = <on or off>
```

where the default is "on".

The information needed for the statistical interpretation of an analysis recast is given in the .info XML file shipped with that analysis, which has to be located in the same directory

---

[2]Covariance matrices can also be used in GAMBIT's COLLIDERBIT [14]; a ColliderBit interface to PYHF is under development, as is the usage of correlation information in CHECKMATE [15].

as the analysis C++ files. For analyses implemented in the PAD (PADForSFS) format, this consists of the PAD/Build/SampleAnalyzer/User/Analyzer (PADForSFS/Build/SampleAnalyzer/User/Analyzer) directory [3, 16]. The .info XML file specifies for each SR the observed number of events <nobs>, the number of expected Standard Model events <nb> and the associated $1\sigma$ uncertainty <deltanb>; the latter might be split into its statistical and systematic components: <deltanb_stat> and <deltanb_sys> [17]. This has to be extended by the appropriate information about the JSON-serialised statistical model or covariance matrix as explained in the following.

## 2.1 Usage of JSON-serialised HISTFACTORY models: interface to PYHF

In order to employ the statistical model information embedded in JSON files provided by the ATLAS collaboration, the analysis' .info file needs to be extended by <pyhf> blocks. These specify the filename(s) of the JSON file(s) and the associated channels and region names within the analysis. In the PYHF language, a channel refers to an ordered ensemble of signal regions treated correlatively in the statistical model. The structure of such a <pyhf> block reads:

```
<pyhf id="Global">
  <name>analysisID.json</name>
  <regions>
    <channel name="Channel1">Region1_1 Region1_2 Region1_3</channel>
    <channel name="Channel2">Region2_1 </channel>
    <channel name="Channel3" is_included="True"> </channel>
    <channel name="Channel4" is_included="False"> </channel>
  </regions>
</pyhf>
```

The id of a <pyhf> block is the label of the corresponding exclusion limit calculation, that is further propagated to the output file to present the results (see below). For analyses involving a single JSON file for all SRs, we recommend setting this label to "Global"; for analyses with multiple JSON files (for the combination of different subsets of SRs), each <pyhf> block needs to be assigned a unique identifier.

The following child block <name> declares the name of the JSON file. This follows the same naming convention as that employed for the analysis implementation, with an optional extra identification suffix if needed.[3] This JSON file needs to be located at the same path as the MADANALYSIS 5 information file. For the ATLAS analyses discussed in section 3, this is automatically realised after the installation of a local version of the MADANALYSIS 5 Public Analysis Database through the MADANALYSIS 5 command line interface (install PAD or install PADforSFS).

In the next child of the <pyhf> block, regions are collected into multiple channels. Each channel includes the name of the recast signal regions corresponding to that specific channel. The channel names must correspond to those used in the JSON file, while the Region<i_j> names have to match the region names chosen in the recast implementation. Usually this pertains to SRs, although control and validation regions could be recast as well. The ordering of the regions is crucial and needs to follow that of the JSON file given by the ATLAS collaboration. Each channel can have a different number of regions, but it must match the available number of regions given in the JSON file. A concrete example from the ATLAS-SUSY-2018-31 [18] analysis, which has three sets of disjoint SRs (regions A, B and C), is [19]

---

[3]For instance, such suffixes can be used to distinguish between full and simplified JSON files, or JSON files combining different subsets of SRs.

```
<pyhf id="RegionA">
  <name>atlas_susy_2018_31_SRA.json</name>
  <regions>
    <channel name="SR_meff">SRA_L SRA_M SRA_H</channel>
    <channel name="VRtt_meff"></channel>
    <channel name="CRtt_meff"></channel>
  </regions>
</pyhf>
```

and analogously for regions B and C. Since validation regions (VR) and control regions (CR) are not included in the recast code, the corresponding channels are left empty.

Upon execution, MADANALYSIS 5 patches the region counts onto the JSON file, thus creating a JSON patchset for the particular BSM hypothesis. The latter is then evaluated through a call to PYHF. More concretely, by patching the signal yields to background samples, MADANALYSIS 5 creates a dynamic statistical model which is then used to compute $p$-values and test statistics for a single parameter of interest, the signal strength $\mu$. Expected and observed limits on the cross section (in pb) are determined via optimising the signal strength to find the 95% confidence level (CL) upper limit, $\mu_{\mathrm{UL}}$. The $1-\mathrm{CL}_s$ value, on the other hand, is computed by setting $\mu = 1$. For details, see [20].

By default, the creation of the patchset includes only channels with at least one region; empty channels (which do not have any region) are removed from the statistical model. Typically this concerns control and validation regions, like for the ATLAS-SUSY-2018-31 example above. So far, MADANALYSIS 5 recast codes do not emulate these regions, because they are not supposed to be sensitive to any BSM signal.[4] Removing them from the statistical model gives a good enough approximation for most purposes and considerably reduces computation time. In some cases however (for instance when the correct statistical evaluation requires a combined fit to signal and control regions), it can be relevant to keep a given channel even if it is not reproduced in the recast code. This is achieved with the `is_included` attribute in the `<channel>` element. If present, it informs MADANALYSIS 5 whether or not to include a given channel while forming the statistical model, overriding the default behaviour. If `is_included` is set to `"True"` for an empty channel, it is hence included in the likelihood calculation assuming that the BSM signal yields in all its bins are zero. An example where this matters is the electroweakino (EW-ino) search ATLAS-SUSY-2018-06 [21], which has two signal and two control regions and whose `<pyhf>` block in the `atlas_susy_2018_06.info` file reads [22]

```
<pyhf id="Global">
  <name>atlas_susy_2018_06_simplified.json</name>
  <regions>
    <channel name="SRlow_cuts"> SR_low </channel>
    <channel name="SRISR_cuts"> SR_ISR </channel>
    <channel name="CRlow_cuts" is_included="True"> </channel>
    <channel name="CRISR_cuts" is_included="True"> </channel>
  </regions>
</pyhf>
```

This is also an example of an analysis recast which makes use of a simplified statistical model (derived with the SIMPLIFY tool from the full statistical model), as indicated by the `_simplified` suffix in the JSON filename. We come back to this feature in section 3.

---

[4]It will, however, be good to include them in the future (whenever feasible) in order to be able to check possible signal contamination in CRs and/or to study cross-analysis correlations.

We now turn to the format of the output file generated by MADANALYSIS 5. Whenever the exclusion is computed by means of the PYHF package, the results are reported in the `CLs_output_summary.dat` file in the form

```
<set> <tag> <SR> <best?> <exp> <obs> <CLs> ||
```

just after the results for individual SRs. The successive elements are the dataset name `<set>`, the analysis name `<tag>`, the description of the subset of combined SRs `<SR>` that contains an explicit `[pyhf]` tag, the flag for best combination (0 or 1), the expected and observed cross section upper limits at 95% CL, and finally the exclusion level, $1 - CL_s$; see [3] for details. No statistical error information is printed (to the right of the double bars), as it is already accounted for in the likelihood calculation. A concrete example reads (values rounded for space reasons)

```
smpl atlas_susy_2018_31 [pyhf]-RegionA-profile  1  0.0016  0.0011 0.9787 ||
```

where `smpl` stands for the identifier of the given event sample, and where a `[pyhf]` tag identifies the combined result. If there is only one likelihood profile, it is always identified as the "best" combination. If there are several region combinations, the one with the lowest expected limit on the cross section is flagged as the "best" one.

A comment is in order regarding the meaning of "expected" limit. PYHF by default reports post-fit (or "aposteriori") expected values, *i.e.* limits after a fit of the background expectations to the observed data.[5] This differs from the usual definition of expected limits in MADANALYSIS 5, in which the observed numbers of events in each SR are set equal to the number of expected background events [3], and which we here call "apriori" expected limits. To ensure consistency between the individual and combined SRs results, MADANALYSIS 5 can now compute both "apriori" and "aposteriori" expected limits. The choice is done via the command

```
set main.recast.expectation_assumption = <apriori or aposteriori>
```

The default is "`apriori`", in which case expected limits from PYHF are determined from a patchset in which the observed numbers of events in each SR are replaced by the number of expected background events. In contrast, when setting the expected-limit computation to "`aposteriori`", the default PYHF output is taken for the combined result, while for individual SR results the numbers of background events are set equal to the observed numbers of events.

## 2.2 Usage of simplified likelihoods through covariance matrices

For the combination of SRs via the simplified likelihood approach [5], we adopted the implementation in SMODELS [12], *i.e.* its PYTHON module `simplified_likelihood.py`, for use in MADANALYSIS 5. In order to comply with the MADANALYSIS 5 framework [3], the covariance information has to be included in the `.info` XML file associated with the analysis recast code. For each SR, the covariance with every other SR can be supplied. There is thus a list of covariances with two entries: the paired regions and the value of the associated covariance. The (self-explanatory) new standard syntax of the `.info` file reads:

```
<analysis id="analysis name" cov_subset="Global">
    <lumi>...</lumi>
    <region type="signal" id="region name">
        <nobs> ... </nobs>
        <nb> ... </nb>
        <deltanb_stat> ... </deltanb_stat>
```

---

[5]For details, see the PYHF discussions #1367 and #1619 on GITHUB.

```
            <deltanb_syst> ... </deltanb_syst>
            <covariance region="first SR name">...</covariance>
            <covariance region="second SR name">...</covariance>
            ...
            <covariance region="last SR name">...</covariance>
        </region>
        ...
</analysis>
```

This specifies, for each SR, the number of observed events `<nobs>`, expected background events `<nb>`, their statistical (`<deltanb_stat>`) and systematic (`<deltanb_syst>`) uncertainties,[6] as well as the covariance matrix elements linking the current region to other regions.

If a covariance element is not supplied, it is considered as a zero entry in the covariance matrix. In addition, if a region does not contain any covariance field, the region itself is omitted from the combination. This feature can be useful if the covariance matrix is available only for a subset of SRs, like for example in the CMS multilepton plus missing transverse energy search CMS-SUS-16-039 [23], which provides covariances only for the 44 SRs of type A out of a total of 158 SRs (see table 1). The above format also enables the use of multiple covariance matrices in the same analysis (for individually combining subsets of SRs) as in, *e.g.*, CMS-SUS-16-048 [24].[7] In order to keep trace of which subset of SRs is combined, we introduce a `cov_subset` attribute, by which users can provide a brief description of the subset of SRs to which the covariance matrix applies. If there is only one covariance matrix, `cov_subset` can conveniently be specified in the `<analysis>` tag. As in section 2.1, we recommend the label "Global" if all SRs are combined. In case of multiple covariance matrices, the `cov_subset` attributes are set directly within the `<covariance>` tags. For the CMS-SUS-16-039 example, we thus have [28]:

```
<analysis id="cms_sus_16_039" cov_subset="SRs_A">
```

and in the case of the CMS-SUS-16-048 analysis, we have [29]:

```
<analysis id="cms_sus_16_048">
  <lumi>35.9</lumi>
  <region type="signal" id="Ewkino_lowMET_M_4to9">
    <nobs>2</nobs>
    <nb>3.5</nb>
    <deltanb>1.0</deltanb>
    <covariance region="..." cov_subset="Ewkino">1.29</covariance>
    <covariance region="..." cov_subset="Ewkino">0.33</covariance>
   ...
  <region type="signal" id="stop_lowMET_PT_5to12">
    <nobs>16</nobs>
    <nb>14.0</nb>
    <deltanb>2.3</deltanb>
    <covariance region="..." cov_subset="stop">6.09</covariance>
    <covariance region="..." cov_subset="stop">4.71</covariance>
    ...
</analysis>
```

---

[6]Instead of `<deltanb_stat>` and `<deltanb_syst>`, which are added in quadrature on run time, it is also possible to give the total uncertainty using the tag `<deltanb>` [17].

[7]This is also used in the recast implementation [25, 26] of the CMS disappearing tracks search CMS-EXO-19-010 [27] for the combination of statistically independent datasets from different years.

The results from the simplified likelihood combination are printed in the output file `CLs_output_summary.dat` in the form

```
<set> <tag> <cov_subset> <best?> <exp> <obs> <CLs> ||
```

analogous to the output format described in section 2.1. The successive elements are the dataset name, the analysis name, the description of the subset of combined SRs (with an `[SL]` prefix indicating that SR combination is performed via the simplified likelihood approach), the flag for best combination (0 or 1), the expected and observed cross section upper limits at 95% CL, and finally the exclusion level, $1 - CL_s$. A concrete example reads

```
defaultset  cms_sus_16_039  [SL]-SRs_A  1  10.4852  11.1534  0.9997 ||
```

The statistical error, usually provided after the double bar, is not printed as it is already encoded in the simplified likelihood calculation. For the expected limits, "`apriori`" and "`aposteriori`" options are available as explained in the previous subsection.

## 3 Included analyses, validation

Signal region combination is currently available for four ATLAS analyses [18,21,30,31] (recast codes [19,22,32,33]) and five CMS analyses [23,24,34–36] (recast codes [28,29,37–40]) in MADANALYSIS 5. An overview is given in table 1. They are included in any local installation of the Public Analysis Database (PAD) of MADANALYSIS 5, achieved via the commands `install PAD` and `install PADForSFS`. In this section, we briefly describe these analyses and illustrate how SR combination improves the quality of the reinterpretation. To this end, we compare mass limits obtained in the best-SR, simplified likelihood and/or full likelihood approaches to the official limits from the ATLAS or CMS collaborations for specific simplified model scenarios used in the experimental publications.

The tool chain that we use for Monte Carlo event simulation is as follows. The hard scattering processes relevant for the investigated simplified models are simulated with MAD-GRAPH5_AMC@NLO (MG5AMC) version 2.6.5 [41]. Leading-order (LO) matrix elements are generated from the built-in `MSSM_SLHA2` model implementation [42] for the SUSY processes considered, and from the public model file [43] for the $t$-channel dark matter example. For each process, we convolute the LO matrix element with the LO set of NNPDF 2.3 parton distribution functions [44, 45], and we generate 200,000 signal events per sample point to limit Monte Carlo uncertainties. PYTHIA version 8.240 [46] is used to handle unstable particle decays, parton showering, and hadronisation; the emulation of detector effects is done either with DELPHES 3 [47] or the SFS framework [16], depending on the specification of each recast analysis. All SUSY particles that do not appear in the simplified model considered are assumed to be decoupled.

Finally, for a 1:1 comparison with the "official" ATLAS and CMS limits, the LO cross sections from MG5AMC are re-scaled to the reference cross sections tabulated on [48] and used by the collaborations; these tabulated cross sections have been obtained with the NNLL-FAST [49–51] and RESUMMINO [52–54] programs, that provide the most precise predictions for SUSY total rates. Simplified statistical models are derived from the full ones provided by the ATLAS collaboration, by means of SIMPLIFY [11] version 0.1.10. The PYHF version employed in this work is 0.6.3.

**ATLAS-SUSY-2018-31 [18]:** This is a search for SUSY in final states with multiple $b$-jets and missing transverse energy. It specifically targets sbottom pair production, $pp \to \tilde{b}\tilde{b}^*$, followed by the cascade decay $\tilde{b} \to b\tilde{\chi}_2^0 \to bh\tilde{\chi}_1^0$. The produced Higgs bosons are assumed to further

decay into a pair of possibly boosted $b$-tagged jets. The analysis has 8 SRs grouped into three classes (regions A, B, C), which target mass spectra of different levels of compression. It was the first one to publish its full statistical model on HEPDATA [55], and was used as the showcase in [7].

The analysis is implemented in MADANALYSIS 5 within the SFS framework [16]. We here use version 2.0 of the implementation [19], which is compliant with the syntax introduced in section 2.1. A detailed description and validation are given in [56].[8]

Figure 1 shows the observed (left panel) and expected (right panel) 95% CL exclusion limits obtained with MADANALYSIS 5 for the $pp \to \tilde{b}\tilde{b}^*$, $\tilde{b} \to b\tilde{\chi}_2^0 \to bh(\to b\bar{b})\tilde{\chi}_1^0$ scenario in the $(m_{\tilde{b}_1}, m_{\tilde{\chi}_2^0})$ plane, with $m_{\tilde{\chi}_1^0}$ fixed to 60 GeV. Following [18], the branching ratios of the $\tilde{b} \to b\tilde{\chi}_2^0$ and $\tilde{\chi}_2^0 \to h\tilde{\chi}_1^0$ decays are set to 100%. The recast limits are computed in three different approaches: using only the best SR (solid red lines), combination of SRs with the full statistical model (solid green lines), and combination of SRs with a simplified statistical model derived with the SIMPLIFY tool (dashed orange lines). These have to be compared to the official limits from ATLAS (in blue); in case of the *observed* exclusion, the error bands indicated for the official limits represent the $1\sigma$ theory uncertainty on sbottom-pair production, while in case of the *expected* exclusion, they represent the $1\sigma$ experimental uncertainty. In the top row of the figure, we make use of signal LO rates as returned by MG5AMC, whereas in the bottom row the results are rescaled to approximate next-to-next-to-leading-order matched with soft-gluon resummation at the next-to-next-to-leading-logarithmic accuracy (NNLO$_{\text{approx.}}$+NNLL). The latter correspond to the predictions used by the ATLAS collaboration in its official publication and are taken from [48].

For the expected limits, all three approaches (best SR, full and simplified statistical model) give very similar results and agree well, at the level of about $1\sigma$, with the ATLAS result. For the observed limits, with LO cross sections the best-SR approach somewhat under-excludes, while SR combination gives a result closer to the official limit. Employing the reference cross sections from [48], the agreement becomes almost perfect. (Henceforth, we will show results only for reference cross sections.) In this simplified-model example, the best SR performs very well. However, this need not be the case for more complicated scenarios, in which the signal may be spread to a larger extent over several SRs. The combination of SR therefore ensures a more reliable and robust interpretation than the best-SR approach. We also note that here the simplified statistical model performs very well, at significantly less CPU cost (about one fifth) than the full one. It can therefore be advantageous to use SR combination with the simplified statistical model for this analysis.

**ATLAS-SUSY-2018-04 [30]:**   This analysis is a search for direct stau production in events with two hadronic taus, $pp \to \tilde{\tau}^+\tilde{\tau}^-$, $\tilde{\tau}^\pm \to \tau^\pm\tilde{\chi}_1^0$. Two event selection strategies are considered, respectively focusing on low-mass and high-mass stau production through dedicated triggers [57] and selections on the missing transverse energy $\not{E}_T$ and the stranverse mass $m_{T2}$ of the final-state system [58, 59].

The MADANALYSIS 5 implementation [32] of the analysis uses DELPHES 3 for the simulation of the ATLAS detector; a detailed description and validation are given in [60]. The combination of SRs through PYHF, using either the full statistical model available from HEPDATA [61] or its simplified version derived with SIMPLIFY 0.1.10, is enabled from version 4.0.

The effect of SR combination is illustrated in figure 2 for the case $pp \to \tilde{\tau}^+_{L,R}\tilde{\tau}^-_{L,R} \to \tau^+\tilde{\chi}_1^0\,\tau^-\tilde{\chi}_1^0$, with the contributions of the mass-degenerate left- and right-chiral staus summed over. The meaning of the various contours is the same as in figure 1. For the expected limit, shown in the right panel, SR combination based on the full statistical model reproduces very well the official ATLAS result. The simplified statistical model, however, gives an over-estimation of

---

[8]All validation notes are also available on the PAD homepage.

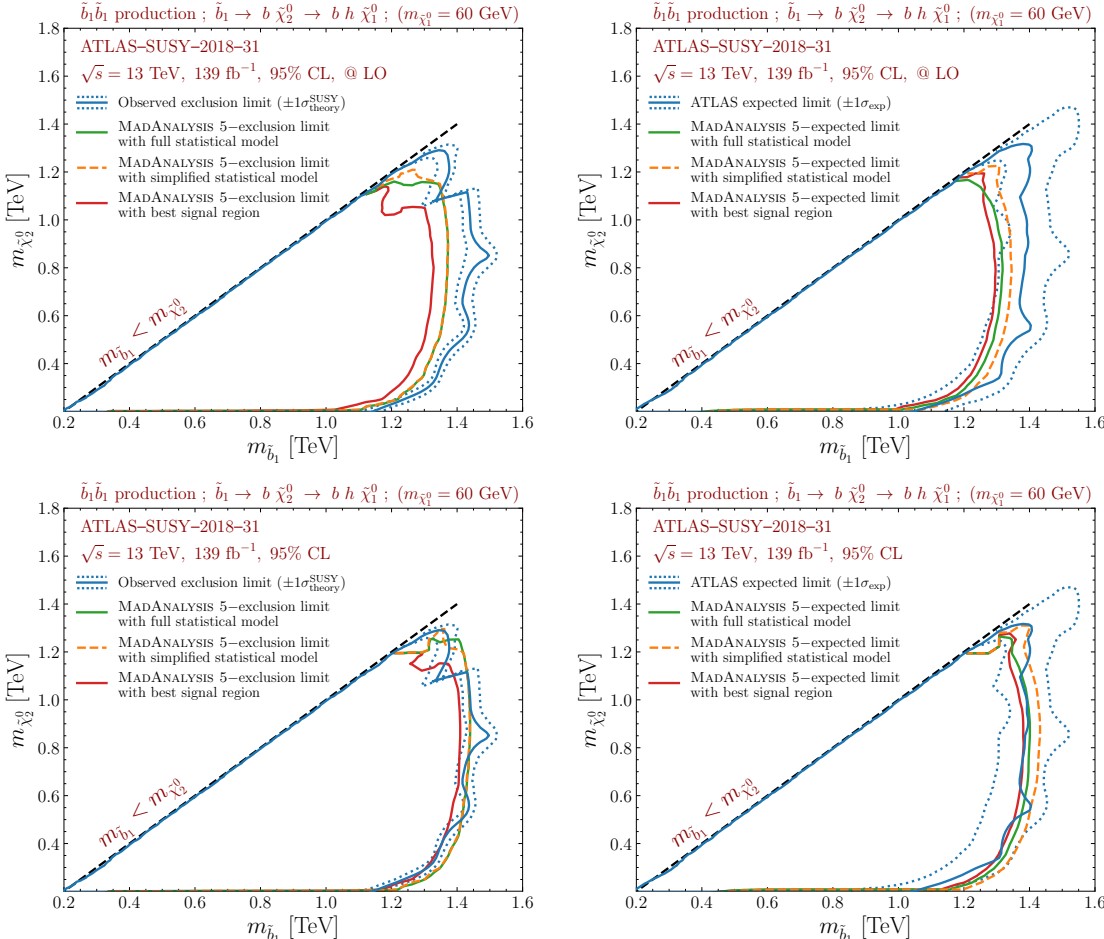

Figure 1: 95% CL exclusion contours for the $pp \to \tilde{b}\tilde{b}^*$, $\tilde{b} \to b\tilde{\chi}_2^0 \to bh\tilde{\chi}_1^0$ simplified model in the $(m_{\tilde{b}_1}, m_{\tilde{\chi}_2^0})$ plane, derived from the ATLAS-SUSY-2018-31 analysis. The left panels show the observed limits and the right panels the expected limits, in the upper row for LO cross sections, in the lower row using the tabulated reference cross sections from [48]. We compare the official limits from ATLAS (blue) to those obtained with MADANALYSIS 5 when using the best signal region only (red), the full statistical model provided by the ATLAS collaboration (green) and an approximate version of it derived with the SIMPLIFY tool (orange).

the sensitivity. Using the best-SR only also turns out to be slightly too aggressive, though the difference to the official expected exclusion line is within $1\sigma$ of the experimental uncertainty. For the observed limit, shown in the left panel, we observe an over-exclusion with all three approaches, although the full statistical model again performs best. This difference originates from the recasting procedure and was also noted in [60], where it was traced to a difference of up to 50% in the effect of the $m_{T2}$ cut (based on the cutflows for two benchmark points with stau masses of 120 and 280 GeV provided by the ATLAS collaboration). We must note here, however, that the analysis involves several identification and reconstruction efficiencies, which are specified only approximately in the ATLAS publication [30]. These efficiencies are used in the recast code to incorporate the multi-level tau tagging, which is not directly possible in DELPHES 3. A fudge factor of 0.7 (reducing the final weights by 30%) would bring the observed limit from MADANALYSIS 5 in agreement with the official ATLAS one. For the expected limit, we note that the MADANALYSIS 5 results shown in the right panel of figure 2 are pre-fit, while

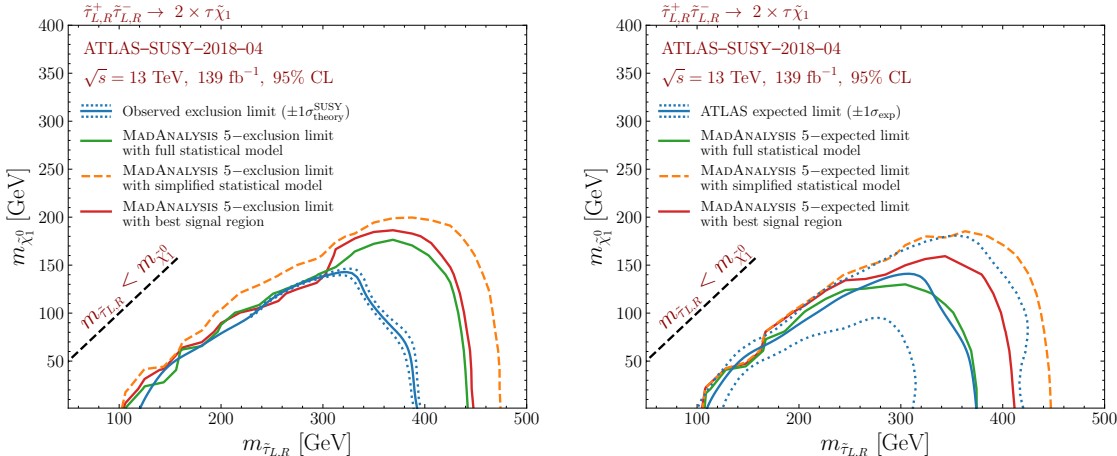

Figure 2: 95% CL exclusion contours as in figure 1, but for the $pp \rightarrow \tilde{\tau}_{L,R}^{+}\tilde{\tau}_{L,R}^{-} \rightarrow \tau^{+}\tau^{-}\tilde{\chi}_1^0\tilde{\chi}_1^0$ simplified model in the $(m_{\tilde{\tau}_{L,R}}, m_{\tilde{\chi}_1^0})$ plane and derived from the ATLAS-SUSY-2018-04 analysis.

the ATLAS expected limit curve seems to be post-fit. The difference between pre-fit and post-fit background numbers (*cf.* last paragraph of section 2.1) turns out to compensate the higher acceptance×efficiency values from the recast code. In any case, it is recommended to use the full statistical model for this analysis.

**ATLAS-SUSY-2018-06 [21]:** This search investigates an electroweakino signal made of three leptons plus $\not{E}_T$ by means of the recursive jigsaw reconstruction technique [62,63]. It specifically targets the production of (wino-like) charginos and neutralinos that further decay into $W$ or $Z$ bosons and the lightest neutralino, $pp \rightarrow \tilde{\chi}_1^\pm \tilde{\chi}_2^0 \rightarrow (W\tilde{\chi}_1^0)(Z\tilde{\chi}_1^0) \rightarrow (\ell\nu\tilde{\chi}_1^0)(\ell\ell\tilde{\chi}_1^0)$. The analysis has two SRs, one vetoing jets, and one requiring 1–3 jets from initial-state radiation.

The MADANALYSIS 5 implementation [22] relies on DELPHES 3 and is described and validated in [64]. The interface to PYHF is enabled from version 5.0 of this implementation. Note that, as mentioned in section 2.1, it is important in this case to include also the CRs in the combination. A complication arises from the fact that the full statistical model provided on HEPDATA [65] leads to issues[9] which so far could not be clarified and thus prevent us from using it for physics purposes. The simplified statistical model obtained with SIMPLIFY, however, yields reasonable results. Consequently, only the latter is included in the MADANALYSIS 5 implementation.

Figure 3 shows the observed and expected bounds on the $pp \rightarrow \tilde{\chi}_1^\pm \tilde{\chi}_2^0 \rightarrow (W\tilde{\chi}_1^0)(Z\tilde{\chi}_1^0) \rightarrow (\ell\nu\tilde{\chi}_1^0)(\ell\ell\tilde{\chi}_1^0)$ signal obtained with MADANALYSIS 5 in the $(m_{\tilde{\chi}_1^\pm/\tilde{\chi}_2^0}, m_{\tilde{\chi}_1^0})$ plane together with the official results from the ATLAS collaboration. While the best-SR approach already leads to a good agreement with the official limits, this is improved by the SR combination. Since the latter involves a combined fit to SRs and CRs, this is a case where emulating also the CRs in the recast code would be beneficial.

**ATLAS-SUSY-2019-08 [31]:** This is a search for electroweakinos in final states with one lepton, $\not{E}_T$, and two $b$-jets consistent with the decay of a Higgs boson. Like ATLAS-SUSY-2018-06, it targets the production of a chargino-neutralino pair, but with the $\tilde{\chi}_2^0$ decaying via a Higgs boson: $pp \rightarrow \tilde{\chi}_1^\pm \tilde{\chi}_2^0 \rightarrow (W\tilde{\chi}_1^0)(h\tilde{\chi}_1^0) \rightarrow (\ell\nu\tilde{\chi}_1^0)(b\bar{b}\tilde{\chi}_1^0)$. The analysis comprises 9 SRs grouped into three classes, which focus on different mass splittings between the $\tilde{\chi}_1^\pm/\tilde{\chi}_2^0$ states (assumed

---

[9]See PYHF issue #1320 for details.

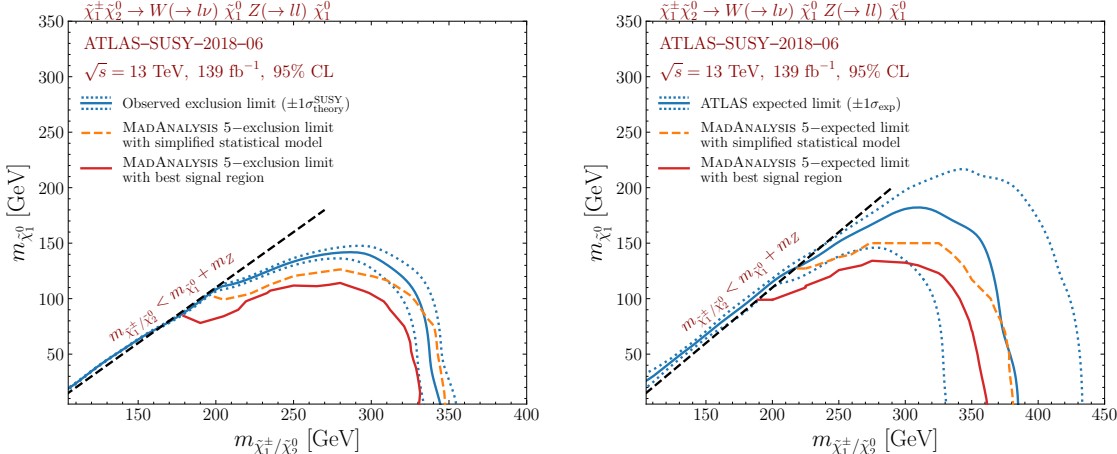

Figure 3: 95% CL exclusion contours as in figure 1, but for the $pp \rightarrow \tilde{\chi}_1^\pm \tilde{\chi}_2^0 \rightarrow (W^\pm \tilde{\chi}_1^0)(Z\tilde{\chi}_1^0) \rightarrow (\ell\nu\tilde{\chi}_1^0)(\ell\ell\tilde{\chi}_1^0)$ simplified model in the $(m_{\tilde{\chi}_1^\pm/\tilde{\chi}_2^0}, m_{\tilde{\chi}_1^0})$ plane and derived from the ATLAS-SUSY-2018-06 analysis.

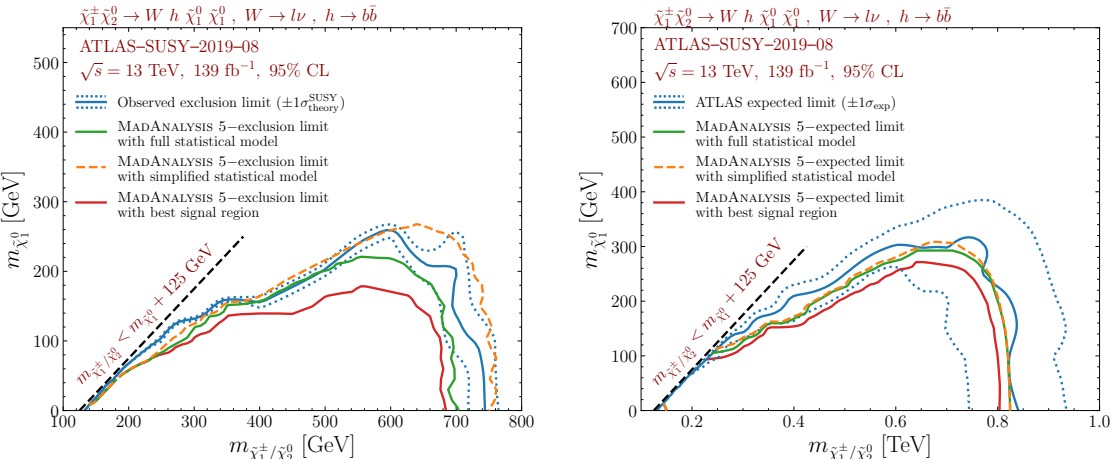

Figure 4: 95% CL exclusion contours as in figure 1, but for the $pp \rightarrow \tilde{\chi}_1^\pm \tilde{\chi}_2^0 \rightarrow (W^\pm \tilde{\chi}_1^0)(h\tilde{\chi}_1^0) \rightarrow (\ell\nu\tilde{\chi}_1^0)(b\bar{b}\tilde{\chi}_1^0)$ simplified model in the $(m_{\tilde{\chi}_1^\pm/\tilde{\chi}_2^0}, m_{\tilde{\chi}_1^0})$ plane and derived from the ATLAS-SUSY-2019-08 analysis.

to be degenerate in mass) and the lightest neutralino $\tilde{\chi}_1^0$, and it relies on various high-level kinematic variables including the contranverse mass of the two $b$-jets [66, 67].

The MADANALYSIS 5 implementation [33] of this analysis makes use of DELPHES 3 for the simulation of the ATLAS detector. We refer to [68] for details and validation information. We use version 6.0, which allows for SR combination through the PYHF package, both on the basis of the full statistical model available from HEPDATA [69] and on that of a simplified one derived with SIMPLIFY [11].

The limits in the $(m_{\tilde{\chi}_1^\pm/\tilde{\chi}_2^0}, m_{\tilde{\chi}_1^0})$ plane obtained with the MADANALYSIS 5 recast for the process $pp \rightarrow \tilde{\chi}_1^\pm \tilde{\chi}_2^0 \rightarrow (W^\pm \tilde{\chi}_1^0)(h\tilde{\chi}_1^0)$ are shown in figure 4, and compared to the official ones from the ATLAS collaboration. Signal region combination with the full statistical model clearly improves the agreement with the ATLAS result as compared to the best-SR approach. The simplified statistical model, on the other hand, performs less well and leads to an over-exclusion. The difference however remains at the level of $1\sigma$ of the theory uncertainty. Overall it is recommended to use the full statistical model for this analysis.

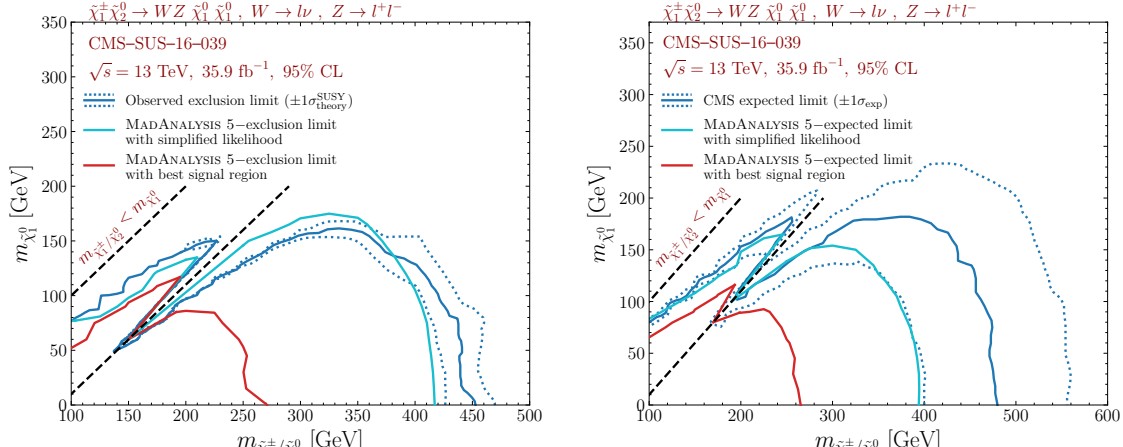

Figure 5: 95% CL exclusion contours for the $pp \to \tilde{\chi}_1^{\pm} \tilde{\chi}_2^0 \to (W \tilde{\chi}_1^0)(Z \tilde{\chi}_1^0) \to (\ell \nu \tilde{\chi}_1^0)(\ell\ell \tilde{\chi}_1^0)$ simplified model in the $(m_{\tilde{\chi}_1^{\pm}/\tilde{\chi}_2^0}, m_{\tilde{\chi}_1^0})$ plane, derived from the CMS-SUS-16-039 analysis. The left panel shows the observed limits, while the right panel shows the expected limits. We compare the official limits from the CMS collaboration (blue) to those obtained with MADANALYSIS 5 when using the best signal region only (red) and when combining SRs with the publicly available covariance matrix information (teal).

**CMS-SUS-16-039 [23]:** This analysis targets the production of charginos and/or neutralinos that decay into final states comprising two or more leptons. It contains 158 SRs defined in terms of $\not{E}_T$, the number and flavours of the leptons, their electric charges and other properties of the produced multi-leptonic system like its invariant or transverse mass.

The implementation of this analysis in MADANALYSIS 5 relies on DELPHES 3 for the simulation of the CMS detector. Details and validation information are available from [70]. We make use of its version 3.0 [28] that includes the covariance matrix for the 44 three-lepton SRs (SRA01–SRA44) available on the analysis twiki page. This class of SRs is dedicated to final states featuring three non-tau leptons and including at least one opposite-sign same-flavour lepton pair. The correlation information for the other classes of SRs is not publicly available.

Figure 5 presents observed (left panel) and expected (right panel) 95% CL exclusion limits obtained with MADANALYSIS 5 for the process $pp \to \tilde{\chi}_1^{\pm} \tilde{\chi}_2^0 \to (W \tilde{\chi}_1^0)(Z \tilde{\chi}_1^0) \to (\ell \nu \tilde{\chi}_1^0)(\ell\ell \tilde{\chi}_1^0)$ in the $(m_{\tilde{\chi}_1^{\pm}/\tilde{\chi}_2^0}, m_{\tilde{\chi}_1^0})$ plane. This process is the target of SRs of class A, for which the covariance matrix is publicly available. The limits are computed by MADANALYSIS 5 in two different approaches, namely using the best SR (red), and combining SRs in the simplified likelihood approach by means of the covariance matrix (teal). These results are compared to the official limits from the CMS collaboration (blue). As for the analyses above, for observed limits the dashed blue lines indicate the $1\sigma$ theory uncertainty on the signal cross section, whereas for expected limits they indicate the experimental uncertainty.

The difference between the best-SR and combined results is striking. Owing to the fine binning of SRs in the CMS analysis, the hypothesised signal populates several of the analysis SRs, reducing consequently the sensitivity of any single region. Only with a statistical combination can the CMS mass limits be reproduced to a good approximation. We conclude that SR combination greatly ameliorates the reinterpretation of the results of this analysis. It would hence be great if covariance matrices were available also for the other 13 classes of SRs of this analysis.

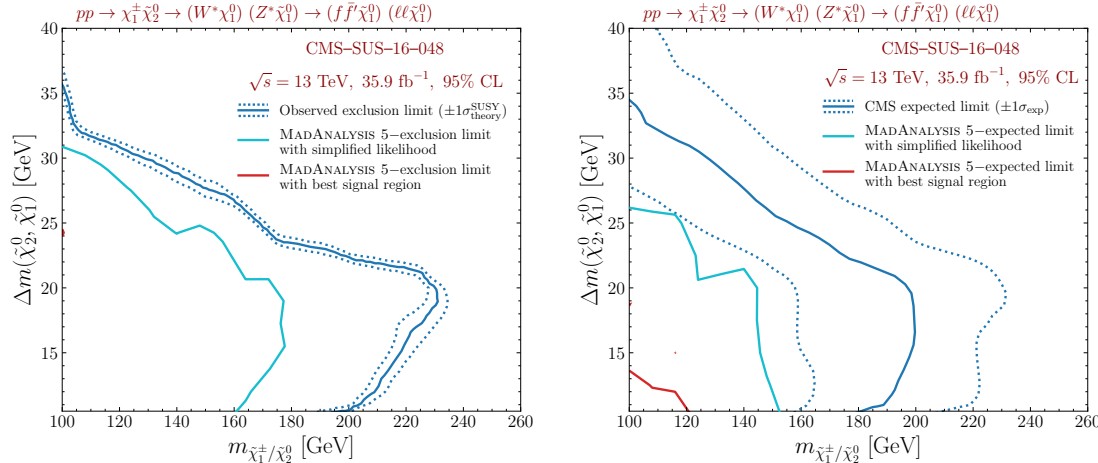

Figure 6: 95% CL exclusion contours as in figure 5, but for the $pp \to \tilde{\chi}_1^\pm \tilde{\chi}_2^0 \to f \bar{f}' \tilde{\chi}_1^0 \, \ell\ell\tilde{\chi}_1^0$ simplified model in the $(m_{\tilde{\chi}_1^\pm/\tilde{\chi}_2^0}, \, \Delta m = m_{\tilde{\chi}_1^\pm/\tilde{\chi}_2^0} - m_{\tilde{\chi}_1^0})$ plane and derived from the CMS-SUS-16-048 analysis.

**CMS-SUS-16-048 [24]:** This CMS search focuses on a signature with two soft leptons ($\ell = e, \mu$) of opposite electric charge and $\not{E}_T$, as is typical from compressed new physics spectra with a dark matter candidate. It relies on vetoes on a large hadronic activity and cuts on high-level observables built from the lepton properties and the missing transverse energy. Two classes of search regions are defined: an electroweakino class of search regions (12 SRs) through cuts on $\not{E}_T$ and the di-lepton invariant mass $M(\ell\ell)$, and a stop class of search regions (9 SRs) through cuts on $\not{E}_T$ and the lepton transverse momenta $p_T(\ell)$. Two covariance matrices are available on the analysis twiki page, one for each of the two classes of regions.

The analysis has been implemented in MADANALYSIS 5 both in the SFS framework [16] and for use with DELPHES 3. Details on the implementation and validation can be found in [16,71]. Version 3 of the 'SFS implementation' [37] and version 2 of the 'DELPHES 3 implementation' [29] include the covariance matrix information, which allows for the construction of simplified likelihoods associated with subsets of the 12 electroweakino and the 9 stop SRs of the analysis. The material is again taken from the analysis twiki page.

For validation, we show in figure 6 exclusion contours derived with the 'SFS implementation' for the electroweakino simplified model in the plane of the $\tilde{\chi}_1^\pm/\tilde{\chi}_2^0$ mass versus its difference with the $\tilde{\chi}_1^0$ mass. The hypothesised signal is $pp \to \tilde{\chi}_1^\pm \tilde{\chi}_2^0 \to (f \bar{f}' \tilde{\chi}_1^0) (\ell\ell\tilde{\chi}_1^0)$, where the $\tilde{\chi}_1^\pm$ and $\tilde{\chi}_2^0$ decays proceed via off-shell $W$ and $Z$ bosons with branching ratios of 100%. The colour code is the same as in figure 5. As for the CMS-SUS-16-039 analysis above, the limits obtained from the best-SR and simplified likelihood approaches are vastly different. In fact, the best SR alone has very little sensitivity to the scenario under consideration (see the expected limits, right panel in figure 6) and does not exclude any of it (see the observed limits, left panel in figure 6). The combination of SRs in the simplified likelihood approach, on the other hand, allows one to reproduce the official bounds from CMS within $1\sigma$–$2\sigma$ of the experimental uncertainty.

**CMS-SUS-17-001 [36]:** This is a search in final states with two oppositely charged leptons ($\ell = e, \mu$), $b$-jets, and $\not{E}_T$. It targets stop-pair production with the stops decaying directly into $t\tilde{\chi}_1^0$ or into $bW\tilde{\chi}_1^0$ via a chargino, as well as direct dark matter production in association with top quarks through scalar or pseudoscalar mediator exchange. The analysis has 13 SRs defined in terms of $\not{E}_T$ and the transverse mass variables $m_{T2}(b\ell b\ell)$ and $m_{T2}(\ell\ell)$. These are further split into same-flavor and different flavor SRs, totalling 26 SRs. In addition, three aggregate

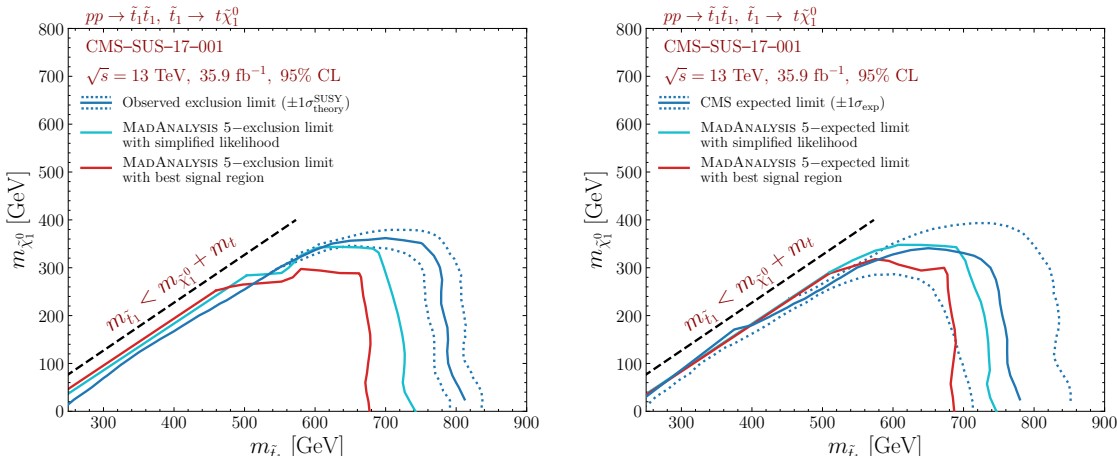

Figure 7: 95% CL exclusion contours as in figure 5, but for the $pp \to \tilde{t}^* \tilde{t} \to (t \tilde{\chi}_1^0)(\bar{t} \tilde{\chi}_1^0)$ simplified model in the $(m_{\tilde{t}}, m_{\tilde{\chi}_1^0})$ plane and derived from the CMS-SUS-17-001 analysis.

SRs are defined in terms of $\not{E}_T$ and $m_{T2}(\ell\ell)$.

The implementation in MADANALYSIS 5 [39], described and validated in [72], is for the three aggregate SRs and relies on DELPHES 3. We use version 3.0 of the code, which employs a prefit covariance matrix for the aggregate SRs constructed from the background uncertainties and correlation matrix given on the analysis twiki page.

Figure 7 shows observed and expected 95% CL exclusion limits for the $pp \to \tilde{t}\tilde{t}^* \to (t\tilde{\chi}_1^0)(\bar{t}\tilde{\chi}_1^0)$ scenario in the $(m_{\tilde{t}}, m_{\tilde{\chi}_1^0})$ plane. The official CMS bounds are reasonably well reproduced, as expected from the mere goal of aggregate regions. Nonetheless, the MADANALYSIS 5 exclusion lines get closer to the official ones when the information from all three aggregate regions is combined. This improvement demonstrates again the importance of using correlation information whenever it is available. Notice also that the correlation between each of the three aggregate regions is about 0.4–0.5; treating them as fully correlated or fully uncorrelated would thus not be correct.

**CMS-SUS-19-006 [34]:** This analysis is dedicated to new physics signals featuring multiple jets and missing transverse energy. It includes 174 SRs defined by the number of reconstructed jets, $b$-tagged jets, the hadronic activity $H_T$ and the amount of $\not{E}_T$. The implementation in MADANALYSIS 5 relies on DELPHES 3; we use its version 6.0 [38] that allows for the combination of all 174 SRs of the analysis. The 12 (overlapping) aggregate regions defined in the analysis are also implemented. Details and validation material are available from [73].

CMS-SUS-19-006 generically targets gluino, stop, sbottom, and squark production. In figure 8, we consider the $pp \to \tilde{g}\tilde{g} \to (t\bar{t}\tilde{\chi}_1^0)(t\bar{t}\tilde{\chi}_1^0)$ scenario for validation. Whereas limits obtained by considering the best-SR approach are too conservative (under-estimating the limit on the gluino mass by about 10%), approximate likelihoods built from the covariance matrix (published on HEPData [74]) allow us to get an agreement with the CMS official results at $1\sigma-2\sigma$. For example, at $m_{\tilde{\chi}_1^0} = 100$ GeV, the observed limit on the gluino mass improves from 1950 GeV in the best-SR approach to about 2260 GeV with combined SRs, to be compared to the official CMS limit of 2180 GeV. Despite the small over-exclusion with combined SRs, this improves the reinterpretation potential of this analysis. We note, however, that for the expected limit, the over-exclusion with combined SRs is as important as the under-exclusion with the best SR only.

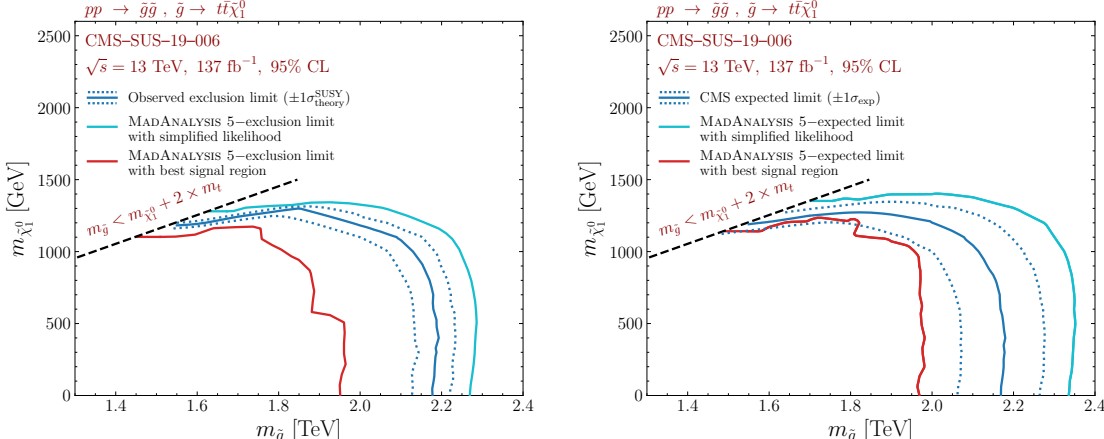

Figure 8: 95% CL exclusion contours as in figure 5, but for the $pp \rightarrow \tilde{g}\tilde{g} \rightarrow (t\bar{t}\tilde{\chi}_1^0)\,(t\bar{t}\tilde{\chi}_1^0)$ simplified model in the $(m_{\tilde{g}}, m_{\tilde{\chi}_1^0})$ plane and derived from the CMS-SUS-19-006 analysis.

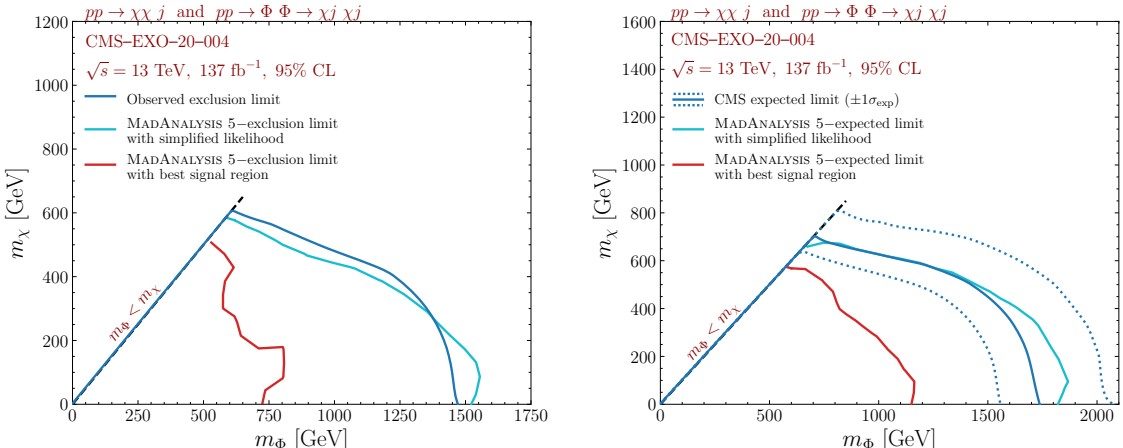

Figure 9: 95% CL exclusion contours as in figure 5, but for a $t$-channel dark matter simplified model ($pp \rightarrow \Phi\Phi \rightarrow \chi j\,\chi j$ and $pp \rightarrow \chi\chi j$) in the $(m_\Phi, m_{\tilde{\chi}})$ plane and derived from the CMS-EXO-20-004 analysis.

**CMS-EXO-20-004 [35]:** This is a search for new particles using events with energetic jets and large missing transverse momentum. Among other signal models, the analysis targets dark matter production in association with at least one highly-energetic jet. A minimum $\not{E}_T > 250$ GeV is required, and separate categories are defined for events with narrow jets from initial-state radiation (mono-jet category) and events with large-radius jets consistent with a hadronic decay of a $W$ or $Z$ boson (low- and high-purity mono-V categories). A shape analysis is then performed of the $\not{E}_T$ spectrum.

The implementation in MADANALYSIS 5 relies on DELPHES 3 and has been provided by the CMS collaboration itself. It consists of the selection for the mono-jet category; a total of 66 SRs are defined, with each of the regions representing one recoil bin in one data-taking year. Information and validation details can be found in [35]; see also the related RAMP seminar by Andreas Albert [75]. We here use version 2.0 of the code [40], which allows for a $\not{E}_T$ shape analysis through the construction of a simplified likelihood based on the yields and the covariance matrix for the SR bins provided on HEPData [76].

For validation, we reproduce in figure 9 the constraints on the fermion portal model

considered in [35]. This is a $t$-channel dark matter signal containing two contributions, namely the production of a pair of scalar mediators $\Phi$ that decay into dark matter $\chi$ and jets ($pp \to \Phi\Phi \to \chi j \, \chi j$) and the direct production of dark matter with an energetic jet ($pp \to \chi\chi j$) through a $t$-channel mediator exchange. Whereas solely very conservative constraints are obtained when relying only on a single $\not{E}_T$ bin (the recast mass limits being a factor of 2 smaller than the official ones), with the shape analysis based on the simplified likelihood the official limits can be reproduced quite well with MADANALYSIS 5. This is explained by the topology of the signal that rarely populates a single $\not{E}_T$ bin, so that a severe loss of sensitivity in the best-SR approach is expected.

## 4 Physics application

In this section, we give an illustrative physics example, that goes beyond the simplified models considered by the experimental collaborations. We start from the simplified model used in ATLAS-SUSY-2018-31 [18], which assumes sbottom pair production with the sbottoms decaying into $b\tilde{\chi}_2^0$ followed by $\tilde{\chi}_2^0 \to h\chi_1^0$. This scenario is not realised in the MSSM for several reasons. First, if the $\chi_1^0$ is bino-like and the $\tilde{\chi}_2^0$ wino-like, as assumed in the simplified model, there is also always a wino-like chargino $\tilde{\chi}_1^\pm$ near in mass to the $\tilde{\chi}_2^0$. Therefore, whenever the mass difference is large enough, $\tilde{b}_1 \to t\tilde{\chi}_1^-$ decays reduce the branching ratio of the $\tilde{b}_1 \to b\tilde{\chi}_2^0$ mode. Second, for sbottoms to decay dominantly via wino-like electroweakinos (instead of directly into the LSP, i.e. $\tilde{b}_1 \to b\tilde{\chi}_2^0$ instead of $\tilde{b}_1 \to b\tilde{\chi}_1^0$), we need $\tilde{b}_1 \sim \tilde{b}_L$. In this case, due to $SU(2)_L$, there is always a $\tilde{t}_1$ more or less near in mass to the $\tilde{b}_1$ with leading decay modes of $\tilde{t}_1 \to b\tilde{\chi}_1^+$, $t\tilde{\chi}_2^0$ and $t\tilde{\chi}_1^0$. Since top quarks decay into $bW$ systems, all these additional modes lead to $b$-rich events, but with somewhat different kinematic features. Moreover, while $\tilde{\chi}_2^0 \to h\tilde{\chi}_1^0$ usually dominates once the neutralino mass difference is large enough, the branching ratio of $\tilde{\chi}_2^0 \to Z\tilde{\chi}_1^0$ is not zero and should be accounted for.

In a realistic MSSM setup, we therefore have a mix of final states originating from

$$pp \to \tilde{b}_1\tilde{b}_1^*, \quad \tilde{b}_1 \to b\tilde{\chi}_1^0, \, b\tilde{\chi}_2^0, \text{ or } t\tilde{\chi}_1^- \quad \text{and} \quad pp \to \tilde{t}_1\tilde{t}_1^*, \quad \tilde{t}_1 \to t\tilde{\chi}_1^0, \, t\tilde{\chi}_2^0, \text{ or } b\tilde{\chi}_1^+. \quad (1)$$

It is therefore interesting to assess how the relevant analyses (in our case ATLAS-SUSY-2018-31 and CMS-SUS-19-006) pick up this signal.

Our benchmark scenario is thus the case where $\tilde{b}_1$, $\tilde{t}_1$, $\tilde{\chi}_1^\pm$, $\tilde{\chi}_2^0$ and $\tilde{\chi}_1^0$ are potentially within LHC reach, while all other supersymmetric particles are heavy, in the multi-TeV range. We call this the T6MSSM scenario.[10] The relevant parameters are the bino and wino masses ($M_1$ and $M_2$), the left squark soft mass for the third generation ($M_{\tilde{Q}_3}$), and the ratio of the two Higgs vacuum expectation values ($\tan\beta = v_2/v_1$).

For definiteness, we fix $M_1 = 60$ GeV and $\tan\beta = 10$ and scan over $M_2$ and $M_{\tilde{Q}_3}$ (with $2M_1 < M_2 < M_{\tilde{Q}_3}$). All other soft masses are set to 5 TeV, $\mu = 1.6$ TeV, and the trilinear couplings $A_{t,b} = -3.5$ TeV (to obtain $m_h \simeq 125$ GeV). This gives a mass spectrum of

$$m_{\tilde{b}_1} \simeq m_{\tilde{t}_1} \simeq M_{\tilde{Q}_3} > m_{\tilde{\chi}_1^\pm} = m_{\tilde{\chi}_2^0} = M_2 > m_{\tilde{\chi}_1^0} = 60 \text{ GeV}. \quad (2)$$

Masses and decay widths are computed with SOFTSUSY 4.1.12 [77,78]. For $M_{\tilde{Q}_3} = 1200$ GeV, we find $m_{\tilde{b}_1} = 1266$ GeV and $m_{\tilde{t}_1} = 1256$ GeV. Figure 10 shows the decay branching ratios as a function of the wino mass parameter $M_2$. As long as these decays are not kinematically suppressed, both the sbottom and the stop decay dominantly via the chargino (which then decays to 100% into $W^\pm\tilde{\chi}_1^0$). In contrast, the decays into the $\tilde{\chi}_2^0$ only have about 20–40% branching ratio. Concretely, we find BR($\tilde{b}_1 \to b\tilde{\chi}_2^0$) $\simeq 0.34$, 0.35, 0.42; BR($\tilde{t}_1 \to t\tilde{\chi}_2^0$) $\simeq 0.32$, 0.31, 0.17; and BR($\tilde{\chi}_2^0 \to h\tilde{\chi}_1^0$) $\simeq 0.93$, 0.84, 0.74 for $M_2 = 200$, 600, 1000 GeV, respectively.

---

[10]This is inspired by the simplified model naming convention in SMODELS.

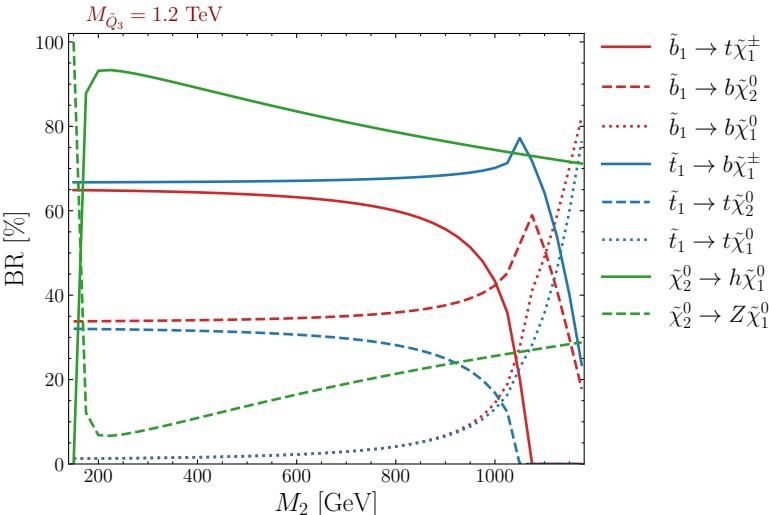

Figure 10: The decay branching ratios in the T6MSSM benchmark scenario as a function of $M_2$, for $M_{\tilde{Q}_3} = 1200$ GeV and $M_1 = 60$ GeV. The decay rate of $\tilde{\chi}_1^\pm \to W^\pm \tilde{\chi}_1^0$ is always 100%.

ATLAS-SUSY-2018-31 excludes sbottoms up to about 1.4 TeV under their peculiar simplified model scenario. CMS-SUS-19-006 excludes sbottoms and stops up to about 1.2 TeV assuming direct decays into a bino LSP, but makes no statement about limits when the decays go via intermediate winos. How the simplified model limits constrain the realistic MSSM case, (1), can easily be checked with SMODELS [12, 13, 79, 80]. Not surprisingly, because of the variety of stop and sbottom decay modes, the constraints on the T6MSSM scenario are rather weak, with the exclusion limit reaching only up to $M_{\tilde{Q}_3} \approx 1$ TeV depending on $M_2$ (and $M_1$ being kept at 60 GeV).[11]

To evaluate the constraints on the T6MSSM scenario with MADANALYSIS 5, we generate $pp \to \tilde{b}_1 \tilde{b}_1^*$ and $pp \to \tilde{t}_1 \tilde{t}_1^*$ events with MG5AMC for a grid of 146 points in the $(M_{\tilde{Q}_3}, M_2)$ plane, with $M_{\tilde{Q}_3} = [1, 1.6]$ TeV and $M_2 = [0.2, 1]$ TeV ($M_1$ being kept at 60 GeV). For each point in the grid, we generate 200,000 hard-scattering events. The events are passed to PYTHIA 8.2 [46] for decays, showering and hadronisation, and then to DELPHES 3 [47] or SFS [16] for the detector simulation in the typical MADANALYSIS 5 recast chain [3], as already explained in section 3. The resulting limits from the ATLAS-SUSY-2018-31 and CMS-SUS-19-006 analyses are presented in figure 11. Here, LO cross sections from MG5AMC are used. As can be seen, the statistical combination of SRs gives a big improvement, excluding stop and sbottom masses up to 1.1–1.3 TeV for wino masses of 0.2–1 TeV. In contrast, using only the best SR, the limits are considerably weaker. The effect is particularly pronounced for the CMS analysis, which has very fine-grained SRs (174 SRs as compared to 8 for the ATLAS analysis); using the 12 aggregate regions of the CMS analysis does not change the picture.

It is relevant to ask whether the limits in figure 11 come mostly from stop or mostly from sbottom production for one or both analyses. To answer this question we pick some points near the exclusion lines and evaluate the constraints on stop and sbottom production separately (*i.e.* considering only $pp \to \tilde{b}_1 \tilde{b}_1^*$ or $pp \to \tilde{t}_1 \tilde{t}_1^*$ events). The result is summarised in table 2. It turns out that the sensitivity to sbottoms and stops is very similar within one analysis. Moreover, we find that a $\geq 95\%$ confidence level exclusion is reached for the points in table 2 only if the total $pp \to \tilde{b}_1 \tilde{b}_1^* + \tilde{t}_1 \tilde{t}_1^*$ production is considered *and* the contributions in all SRs are combined.

---

[11]This is easily understood by considering that, for instance, for BR($\tilde{b}_1 \to b\tilde{\chi}_2^0$) = 0.4 and BR($\tilde{\chi}_2^0 \to h\tilde{\chi}_1^0$) = 0.8, only 10% of sbottom events yield the $bbhh\tilde{\chi}_1^0\tilde{\chi}_1^0$ final state targeted in ATLAS-SUSY-2018-31.

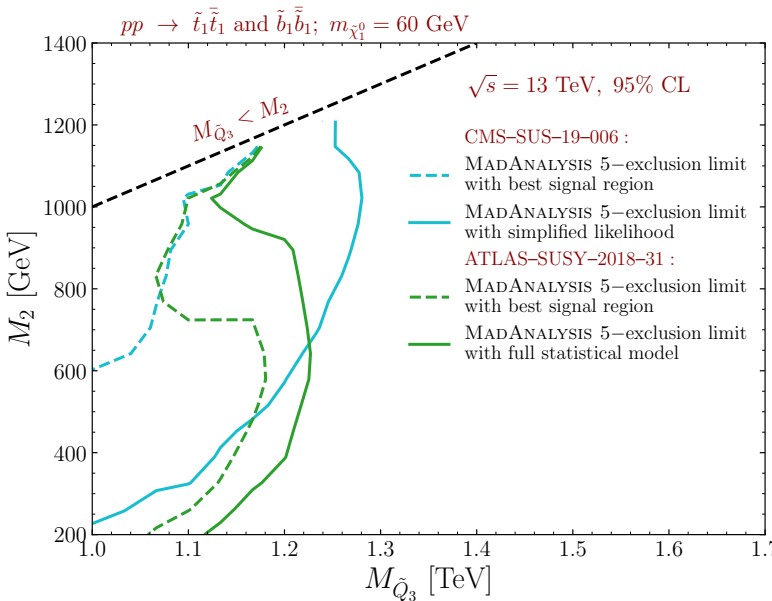

Figure 11: 95% confidence level exclusion limits on T6MSSM scenario in the plane of stop/sbottom *versus* wino mass parameters ($m_{\tilde{t}_1} \simeq m_{\tilde{b}_1} \simeq m_{\tilde{Q}_3}$; $m_{\chi_2^0} \simeq m_{\chi_1^\pm} \simeq M_2$; $m_{\tilde{\chi}_1^0} = M_1 = 60$ GeV). The results for ATLAS-SUSY-2018-31 are in green, the results for CMS-SUS-19-006 in blue. Full lines are the limits with, dashed lines without SR combination.

Since the T6MSSM scenario features wino-like $\tilde{\chi}_1^\pm$ and $\tilde{\chi}_2^0$, complementary constraints come from electroweakino searches. Indeed, as illustrated figure 12, ATLAS-SUSY-2019-08, targeting the $Wh(\to b\bar{b}) + \not{E}_T$ final state, excludes $M_2$ (*i.e.* wino masses) up to about 600 GeV. The same result is obtained from the simplified model limits in SMODELS 2.1. The multi-lepton searches targeting $WZ + \not{E}_T$ final states, on the other hand, provide no relevant constraints, as can also be seen in figure 12. This is no surprise as, with decay branching ratios of $\approx$ 70–90%, $\tilde{\chi}_2^0 \to h\tilde{\chi}_1^0$ decays largely dominate over $\tilde{\chi}_2^0 \to Z\tilde{\chi}_1^0$ decays. Figure 12 also serves to illustrate that the best-SR approach is not always more conservative. Indeed, for $M_2 \lesssim 350$ GeV the CMS-SUS-16-039 best-SR result is too aggressive. This can happen when less events are observed than expected in the best SR, or there are small excesses in other SRs. In any case, the statistical combination of SRs is important for a reliable reinterpretation.

Table 2: $1 - \text{CL}_s$ values from the ATLAS (SUSY-2018-31) and CMS (SUS-19-006) analyses without and with combining SRs, for $M_{\tilde{Q}_3} = 1200$ GeV and three values of $M_2$; the different columns compare sbottom production only ($\tilde{b}_1 \tilde{b}_1^*$), stop production only ($\tilde{t}_1 \tilde{t}_1^*$), and sbottom+stop production taken together (total). Only if the total production is considered and the SRs are combined we obtain $1 - \text{CL}_s \geq 0.95$.

| analysis | method | $M_2 = 600$ GeV | | | $M_2 = 800$ GeV | | | $M_2 = 1$ TeV | | |
|---|---|---|---|---|---|---|---|---|---|---|
| | | $\tilde{b}_1 \tilde{b}_1^*$ | $\tilde{t}_1 \tilde{t}_1^*$ | total | $\tilde{b}_1 \tilde{b}_1^*$ | $\tilde{t}_1 \tilde{t}_1^*$ | total | $\tilde{b}_1 \tilde{b}_1^*$ | $\tilde{t}_1 \tilde{t}_1^*$ | total |
| ATLAS | best-SR | 0.71 | 0.66 | 0.94 | 0.70 | 0.59 | 0.91 | 0.29 | 0.21 | 0.57 |
| | combined | 0.83 | 0.80 | **0.98** | 0.84 | 0.74 | **0.97** | 0.80 | 0.56 | **0.92** |
| CMS | best-SR | 0.31 | 0.37 | 0.62 | 0.38 | 0.45 | 0.73 | 0.29 | 0.38 | 0.70 |
| | combined | 0.79 | 0.71 | **0.96** | 0.89 | 0.83 | **0.99** | 0.93 | 0.82 | **0.99** |

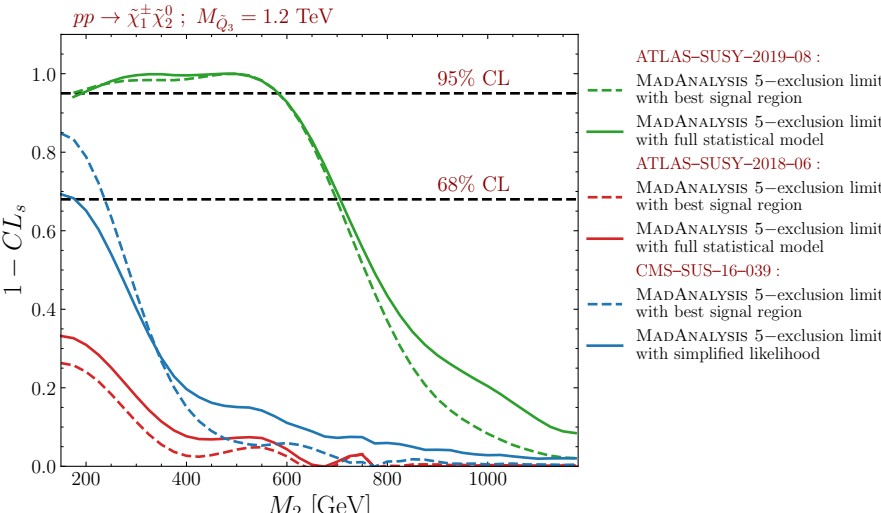

Figure 12: $1 - \mathrm{CL}_s$ values from electroweakino searches as function of $M_2$, for $M_1 = 60$ GeV and $\mu = 1.6$ TeV. Full lines are with, dashed lines without SR combination.

# 5 Conclusions and outlook

We presented the implementation and usage of SR combination in MADANALYSIS 5 via two methods: an interface to the PYHF package making use of statistical models in JSON-serialised format provided by the ATLAS collaboration, and a simplified likelihood calculation making use of covariance matrices provided by the CMS collaboration. Currently, there are recast codes for four ATLAS and five CMS analyses in the MADANALYSIS 5 Public Analysis Database for which this new functionality can be exploited.

We demonstrated the associated gain in physics reach for reinterpretation studies in two ways. First, we reproduced mass limits on simplified model scenarios as published by the ATLAS and CMS collaborations for the analyses considered. Second, we performed a case study of a realistic MSSM scenario in which stops and sbottoms have a variety of decay modes into charginos and neutralinos. Our results show that the statistical combination of disjoint SRs in reinterpretation studies, using more of the data of an analysis, gives more reliable and robust results than the best-SR approach, which uses only the most sensitive SR, for the statistical interpretation of a hypothesised signal.

Next in line of development is the statistical combination of results from different, independent analyses, a functionality that is already available in SMODELS v2.2, and should soon also be adopted in MADANALYSIS 5. Moreover, in order to externalise and make it available to entire HEP community, the SMODELS and MADANALYSIS 5 teams are currently working together to create a universal toolbox for statistics handling in the context of reinterpretation studies.

Before ending this paper, we wish to congratulate the ATLAS collaboration for the big step forward of publishing full statistical models for their analyses on HEPDATA. As discussed in detail in [4], their usefulness goes way beyond the application for SR combinations, which is the topic of this paper. Indeed they open a whole new realm for analysis preservation and reuse. We therefore hope that the publication of full statistical models will continue and soon become standard practice.

# Acknowledgements

We thank the ATLAS and CMS collaborations for making extended information available for their analyses, enabling the reproduction and reuse outside the experimental collaborations. The information on background correlations, which is the basis of this paper, is especially appreciated. We also thank Wolfgang Waltenberger for invaluable discussions and for his engagement in code sharing between SMODELS and MADANALYSIS 5.

**Funding information** This work was supported in part by the IN2P3 master project "Théorie – BSMGA", and by the French Agence Nationale de la Recherche (ANR) under grants ANR-21-CE31-0013 (project DMwithLLPatLHC) and ANR-21-CE31-0023 (PRCI SLDNP). J.Y.A. thanks the LPSC Grenoble and the LPTHE for hospitality and financial support for research visits during the completion of this work.

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
