# Peer review of "Signal region combination with full and simplified likelihoods in MadAnalysis 5"

_SciPost Physics, doi:SciPost Phys. 14, 009 (2023)_

## Round 1 · Referee Report · Anonymous (Referee 1) · 2022-8-2

Strengths

  1. Pedagogical description of the implementation method.
  2. Validation summaries for 9 searches.

Weaknesses

  1. Some of the validation results are problematic in terms of accuracy.

Report

The paper "Signal region combination with the full and simplified likelihoods in MADANALYSIS5" by G. Alguero, J. Araz, B. Fuks, and S. Kraml is a welcome development in the field of recasting at the LHC. It provides a tool, as a part of the MadAnalysis package, for automatic combination of signal regions in the searches published by ATLAS and CMS. Several methods for combination are implemented, depending on the data provided by collaborations. A number or recent ATLAS and CMS searches are making use of the multiple-bin signal regions where the exclusion is decided on the basis of a fit to histograms in distributions of certain kinematic variables. This approach has advantages over the "best signal region" approach which used to be more common in the past (although still used widely). As the authors demonstrate, the method has a profound impact in the CMS searches which define a large number of signal region (~ 100).

In the second section of the paper, the authors provide a detailed description of the implementation of statistical methods and interfaces to various data sources. For the ATLAS searches they use PHYF package and JSON format input files provided by the experiment. For the CMS, they use covariance matrices provided by the collaboration. I find the examples showing the details of implementation particularly useful.

The next section is devoted to validation of the searches. While some of them show a clear advantage, in other cases one can have doubts about their validity. I will specifically list my questions here:
ATLAS-SUSY-2018-31: I am not sure what is the purpose of showing LO analysis. It is just for this search and is clearly off. Not surprisingly as the missing K-factors can easily be of order 1.5. Otherwise the agreement is excellent (but also the gain from the combination rather small).
ATLAS-SUSY-2018-04: The implementation inherits problems of the original implementation, ie. overshooting the acceptance rate by some 30%. What is puzzling to me is why they can reproduce the expected limit and fail with the observed one? I would naively expect a similar level of (dis)agreement. This question equally applies to ATLAS-SUSY-2019-08, CMS-SUS-16-039, CMS-SUS-19-006.
CMS-SUS-16-039: While the agreement in the observed limit is clearly impressive in the high mass region, there is clearly a problem in the 3-body decay region. Is it clear why? In any case the improvement with respect to the best SR is impressive.
CMS-SUS-19-006: The authors downplay the over-exclusion but to me it seems quite dangerous. If I read the plot correctly for the expected limit at 0 LSP mass, this shifts the bound from 2170 GeV to 2350 GeV. This means the upper limit on the cross section is factor 2.4 too strong. Perhaps it does not look that bad in the plot but numerically it is not a negligible number. I would like to see, if possible, how the upper limits on cross section compare in the exclusion plot (at least in a reasonable vicinity of the exclusion line).

The fourth section provides a toy example of a realistic MSSM setup (in contrast to the simplified models from the previous section). The advantage of the improved statistical treatment is clearly demonstrated.

Requested changes

My main request, apart from the minor questions in the main report, is to show how the upper limits on cross section compare between an experiment and a recast. This is certainly more interesting than the LO analysis presented for one of the searches. I am aware that these data are not always available but as far as I remember ATLAS used to provide such information in the auxiliary plots. I think this would would give more confidence on the validity of the presented approach.

  • validity: high
  • significance: top
  • originality: high
  • clarity: top
  • formatting: perfect
  • grammar: perfect

Author:  Sabine Kraml  on 2022-08-31  [id 2776]

(in reply to Report 1 on 2022-08-02)
Category:
answer to question

We thank the referee for the careful assessment of our work. To the questions regarding section 3 (the validation of individual analyses), we reply as follows:

*) ATLAS-SUSY-2018-31, purpose of showing LO analysis: it is not uncommon that (exploratory) phenomenological studies use LO cross sections, as computing higher-order ones is CPU-time consuming and, depending on the model, not always straightforward (e.g., when the available calculations are for certain limits only). Also, in the case of the MSSM, the reference cross sections provided by the LHC SUSY Cross Sections WG are for the particular simplified-model assumptions and do not apply to more general MSSM scenarios. We think it is useful to illustrate the effect in one example. Showing LO results for all analyses discussed in section 3 would clearly be an unwarranted proliferation of plots, but one example can be useful to our mind.

*) ATLAS-SUSY-2018-04, and why we can reproduce the expected limit but fail with the observed one: in fact the agreement of the expected limit curves is somewhat accidental. As explained in the paper, the acceptance\times efficiency values from the MadAnalysis5 implementation of this analysis are roughly 30% higher than those from ATLAS. Therefore, the excluded cross section is smaller, and the observed limit on the stau mass stronger than in the official ATLAS result. For the expected limit, however, this effect is compensated by the fact that we report "apriori" (pre-fit) expected values, while ATLAS seemingly reports "aposteriori" (post-fit) expected values. The difference between pre-fit and post-fit is discussed in the last paragraph of section 2.1. Incidentally, for this particular analysis, the difference between pre-fit and post-fit background numbers is just of the right size to bring the ATLAS and MA5 expected limit curves in agreement. Had we used post-fit expected values, we would see roughly the same over-exclusion as for the observed limit. For better illustration, the attached "supplement.pdf" contains a concrete example.
We have added a remark in the paper (at the end of the last paragraph discussing ATLAS-SUSY-2018-04) to make this clear.

*) CMS-SUS-16-039: why in the 3-body decay region the limit derived with MA5 doesn't match the CMS one. In fact we had not explicitly simulated the 3-body decays with off-shell W and Z bosons. We thought this was clear from the figure caption and plot title, but agree that it is confusing that the limit curves don't match. We have now updated Figure 5 properly including also the off-shell region. The CMS exclusion contour is now reproduced well in both the high mass and the 3-body decay regions.

*) CMS-SUS-19-006, over-exclusion in particular in the expected limit: In reply to the referee's concern, we modified the last sentence of the relevant paragraph (see resubmission letter)

The requested plots showing the ratio of MA5/CMS excluded cross sections, for observed and expected limits, are attached. We agree that a detailed comparison of upper limits on a cross section obtained in a recast and with those provided officially is a very good way to assess the quality of an implementation of any specific recasting program. However, this kind of validation lies outside the scope of the present paper, which only focuses on the statistical modelling. We therefore decided not to include such plots and the related discussion in our manuscript.

As an aside, we note the information provided by CMS is sometimes a mix of pre-fit and post-fit background numbers (e.g., background numbers of events being pre-fit, but the covariance matrix being post-fit, or vice-versa) and the distinction is not always made clear. We are discussing this problem with the CMS collaboration, but do not want to enter into this discussion in the present paper.

Attachment:

supplement.pdf

---

## Round 2 · Referee Report · Anonymous (Referee 1) · 2022-9-8

Report

The authors have addressed my questions and modified the draft adequately.
  • validity: -
  • significance: -
  • originality: -
  • clarity: -
  • formatting: -
  • grammar: -

Author:  Sabine Kraml  on 2022-09-08  [id 2797]

(in reply to Report 1 on 2022-09-08)
Category:
remark

We thank the referee for the speedy assessment (1 week!) and are looking forward to the publication of our paper.

---

## Round 2 · Referee Report · Nishita Desai (Referee 2) · 2022-9-20

Strengths

  1. First public code to use full likelihoods published by ATLAS. Analyses with Simplified likelihood from CMS also have been implemented. This will encourage more experiments and analyses groups to publish this kind of data products.

  2. Clear demonstration of improvement in reach from using combinations of signal regions with respect to best sensitivity methods used before.

Weaknesses

I expect this paper will become a reference for future users to make use of published full likelihoods. In that case, it would be good to have a validation summary (see report).

It would also help if a short appendix for usage of this new functionality can be provided.

I do not see any significant deficiencies otherwise.

Report

This paper describes a comprehensive implementation of using published ATLAS and CMS likelihoods which has been a longstanding desirable for phenomenological studies. It clearly demonstrated how this information, if published by experiments, can be used by the wider community.

The paper is well written and contains detailed implementation notes and explanation of discrepancies when they are seen. However, in places where there is significant discrepancy with published experimental limits, it would be good for the user to quickly know this without having to read through each implementation. In places where MA5 fails conservatively compared to published limits, this is acceptable. However, there are situations (e.g. CMS–SUS–19–006 and ATLAS–SUSY–2018–04) where MA5 seems very aggressive and several sigma in excess of experimental limit. It would be good to include a table with analysis name and topology of each of the searches saying whether the validation is within 1(2) sigma or there is a higher discrepancy and whether it is under or over estimating. It would also be good to have a rule of thumb for where there are known problems with using published likelihoods (i.e. known missing efficiencies that the authors recommend be supplied).

Requested changes

  1. Table of validation summary for each implemented search
  2. Short appendix with user instructions
  3. A line or two in the conclusions about where the authors would like more input from experimentalists that is currently missing and causing difficulties in reinterpretation.

  • validity: top
  • significance: top
  • originality: high
  • clarity: top
  • formatting: excellent
  • grammar: perfect

Author:  Sabine Kraml  on 2022-09-21  [id 2836]

(in reply to Report 2 by Nishita Desai on 2022-09-20)

Thank you for taking the time to review our paper and for the (non-anonymous!) report. We are very pleased by the positive assessment and the high importance and high quality that are accorded to our work. The requested changes are, however, a bit misaligned with the spirit of our paper, and somewhat orthogonal with its main message (so that they may dilute it, which we want to avoid).

1. Table of validation summary for each implemented search

The purpose of our paper is to show the value of detailed information on the statistical modeling from the experiments, not the validation of individual recast codes (which is already published elsewhere). Including a validation summary would, to our mind, draw attention away from the main point of the paper and thus dilute its message and impact.

Apart from this concern, the exact difference to the result derived by the experimental collaboration depends on a number of things, in particular the location in phase space; whether the quantification in terms of sigmas is based on the cut-flows or on the exclusion line*); whether, for any point along the exclusion line, one considers the difference in x or y direction; and whether “1 sigma” includes experimental uncertainties only, or also theory uncertainties. (When including both theoretical and experimental uncertainties, MA5 is nowhere “several sigma in excess of the experimental limit”.) Given these factors of arbitrariness and the above concern about diluting the message of the paper, we wish to refrain from adding a validation summary.

*) Another way to quantify the difference to the official experimental result is the ratio of MA5/official (official=ATLAS or CMS) excluded cross sections; plots of this ratio were requested by the first referee and are available in the supplementary material to version 1 of the submission. However, this cannot be phrased in terms of standard deviations.

2. Short appendix with user instructions (for usage of the new functionality)

There are only two new commands

set main.recast.global_likelihoods = <on or off>

and

set main.recast.expectation_assumption = <apriori or aposteriori>

They are explained in Section 2. The defaults are “on” and “apriori”, so in principle there’s nothing to do for the user to invoke the new functionality, everything is automatic. We do not think that this merits an appendix.

3. A line or two in the conclusions about where the authors would like more input from experimentalists

The conclusions summarize our work and give an outlook to future developments. We could repeat the question regarding pre-fit versus post-fit background numbers and uncertainties (raised in section 3) at the end of the conclusions. However, this would weaken the statement in the last paragraph about the extraordinary scientific value of full statistical models. So we prefer to keep the conclusions as they are, with a very clear message.

For the same reason, we do not think that complaints about too approximate reconstruction efficiencies or missing HEPData entries for some analyses would be appropriate in the conclusions. By the way, regarding a “rule of thumb for where there are known problems with using published likelihoods”: missing efficiencies are a problem on their own, they have nothing to do with the usage of published likelihoods.

For completeness we also want to point out that MadAnalysis5 is not the first public code to use full likelihoods published by ATLAS: while it is the first simulation-based public framework that provides this functionality, the first public tool to do so was SModelS, already two years ago.

---

## Round 2 · Author Response

We thank the referee for the careful assessment of our work. We have taken the comments and questions of the "Anonymous Report 1 on 2022-8-2 (Invited Report)" into account, revised our paper accordingly (minor revision as requested by the editor) and submitted a detailed reply to the referee on https://scipost.org/submissions/2206.14870v1/#report_1

We hope that with these clarifications and small modifications our paper is now suitable for publication in SciPost Physics.

---

## Round 2 · List of Changes

• ATLAS-SUSY-2018-04: added a remark at the end of the last paragraph discussing ATLAS-SUSY-2018-04 to clarify why we can reproduce the expected limit but fail with the observed one:

"For the expected limit, we note that the MadAnalysis5 results shown in the right panel of figure 2 are pre-fit, while the ATLAS expected limit curve seems to be post-fit. The difference between pre-fit and post-fit background numbers (cf. last paragraph of section 2.1) turns out to compensate the higher acceptance x efficiency values from the recast code."

  • CMS-SUS-16-039: updated figure 5 including also the off-shell region in the MadAnalysis5 recast.

  • CMS-SUS-19-006: in reply to the referee's concern regarding the over-exclusion in particular in the expected limit, we modified the last sentence of the relevant paragraph ("Despite the small over exclusion ...") the following way:

"For example, at $m_{\tilde\chi_1^0}=100$~GeV, the observed limit on the gluino mass improves from 1950~GeV in the best-SR approach to about 2260~GeV with combined SRs, to be compared to the official CMS limit of 2180~GeV. Despite the small over exclusion with combined SRs, this improves the reinterpretation potential of this analysis. We note, however, that for the expected limit, the over-exclusion with combined SRs is as important as the under-exclusion with the best SR only."

---

## Editorial Decision

published